# Differences in modifiable cancer risk behaviors by nativity (US-born v. Non-US-born) and length of time in the US

**LaShae D. Rolle**[1]*, **Alexa Parra**[2], **Amrit Baral**[1,2], **Rolando F. Trejos**[3], **Maurice J. Chery**[1], **Reanna Clavon**[4], **Tracy E. Crane**[1,4]

1 Department of Public Health Sciences, University of Miami Miller School of Medicine, Miami, Florida, United States of America, 2 University of Miami School of Nursing and Health Studies, Miami, Florida, United States of America, 3 University of South Florida College of Public Health, Tampa, Florida, United States of America, 4 Sylvester Comprehensive Cancer Center, University of Miami Miller School of Medicine, Miami, Florida, United States of America

* lashaerolle@miami.edu

**Data Availability Statement:** Our study utilized merged datasets from the 2010 and 2015 NHIS obtained via the IPUMS Health Surveys website,

## Abstract

Previous studies have identified racial-ethnic disparities in modifiable risk factors for cancers. However, the impact of US nativity on these risks is understudied. Hence, we assessed the association between US nativity and length of time in the US on modifiable cancer risk factors. Utilizing the 2010 and 2015 National Health Interview Survey datasets, we analyzed 8,861 US-born and non-US-born adults. Key variables included age, sex, race-ethnicity, education, income, diet, body mass index, physical activity, alcohol consumption, and smoking. Statistical methods included descriptive statistics and regression. Most respondents were US-born (n = 7,370), followed by long-term (≥15 years, n = 928), and recent (<15 years, n = 563) immigrants. Moderate-to-vigorous physical activity was higher among US-born individuals (342.45 minutes/week), compared to recent (249.74 minutes/week) and long-term immigrants (255.19 minutes/week). Recent immigrants consumed more fruits (1.37 cups/day) and long-term immigrants more vegetables (1.78 cups/day) than US-born individuals. Multivariate analyses found recent immigrants had lower odds of consuming alcohol (AOR: 0.33, 95% CI: 0.21–0.50) and smoking (AOR: 0.30, 95% CI: 0.19–0.46), and higher odds of meeting fruit consumption guidelines (AOR: 2.80, 95% CI: 1.76–4.45) compared to US-born individuals. Long-term immigrants had lower odds of alcohol consumption (AOR: 0.56, 95% CI: 0.37–0.84) and smoking (AOR: 0.42, 95% CI: 0.30–0.59), and higher odds for meeting fruit (AOR: 1.87, 95% CI: 1.22–2.86) and fiber (AOR: 2.03, 95% CI: 1.02–4.05) consumption guidelines. Our findings illustrate the importance of considering the impact nativity and length of US residency has on health. Our findings underscore the need for culturally tailored public health strategies.

## Introduction

Cancer disparities and inequities in the United States (US), such as differences in cancer incidence, mortality rates, and access to screening and treatment, have resulted in increased cancer

and dietary variables computed using algorithms developed by the National Cancer Institute (NCI) staff to transform screener responses into quantified estimates of individual dietary intake taken from NCI's Epidemiology and Genomics Research Program website (16,17). Blewett LA, Drew JAR, King ML, Williams KCW, Chen A, Richards S, et al. IPUMS Health Surveys: National Health Interview Survey, Version 7.3 [Internet]. Minneapolis, MN: IPUMS; 2023 [cited 2023 Dec 8]. Available from: https://nhis.ipums.org/nhis/ 17. National Cancer Institute. Epidemiology and Genomics Research Program. 2023 [cited 2023 Dec 20]. Epidemiology and Genomics Research Program. Available from: https://epi.grants.cancer.gov/.

**Funding:** The author(s) received no specific funding for this work.

**Competing interests:** The authors have declared that no competing interests exist.

**Abbreviations:** ACS, American Cancer Society; AOR, Adjusted Odds Ratio; BMI, Body Mass Index; CDC, Centers for Disease Control and Prevention; CI, Confidence Interval; MVPA, Moderate-to-Vigorous Physical Activity; NCHS, National Center for Health Statistics; NHIS, National Health Interview Survey; SAS, Statistical Analysis System; SDoH, Social Determinants of Health; US, United States; VIF, Variance Inflation Factor; WCRF/AICR, World Cancer Research Fund/American Institute for Cancer Research.

risk, lower cancer screening rates and other preventive measures, delayed cancer diagnosis, poorer cancer-related treatment outcomes, and increased cancer-related burden [1–3]. While overall cancer rates for most cancer types are declining across racial-ethnic subpopulations [4], discrepancies among different population subgroups remain [5]. Past literature suggests that factors contributing to the creation and continuation of these disparities and inequities in cancer care involve healthcare access and affordability, insurance status, and nativity [4, 6, 7]. Moreover, in the US, socio-cultural and modifiable lifestyle factors play a central role in the overall lifetime risk of a cancer diagnosis and cancer- related outcomes [8, 9].

According to the American Cancer Society (ACS), approximately 18% of all cancers can be attributed to risk factors such as excess body weight, physical inactivity, alcohol consumption, poor nutrition, and smoking cigarettes, highlighting the potential for prevention through lifestyle modifications [10, 11]. The ACS Guideline for Diet and Physical Activity and World Cancer Research Fund/American Institute for Cancer Research (WCRF/AICR) in their Third Expert Report endorse maintaining a healthy body weight, being physically active, limiting tobacco and alcohol consumption, and adhering to a nutritious diet that focuses on a variety of plant foods while limiting intake of red and processed meats, sugar-sweetened beverages, highly processed foods, and refined grain products [11, 12].

The US has the highest number of foreign-born individuals (immigrants) compared to any other country in the world, and the number of new immigrants arriving in the US is steadily increasing [13, 14]. It is estimated that in the next few decades immigrants and their descendants will account for 88% of the US population growth [13, 14]. Foreign-born individuals in the US face health disparities related to social determinants of health (SDoH). SDoH are conditions in the environments where people are born, live, learn, work, play, and age that affect a wide range of health and quality of life outcomes and risks [15]. Examples of SDoH contributing to health disparities in immigrants include limited access to healthcare, socioeconomic status, language barriers, and living conditions [16, 17]. These are often compounded by stigma (negative attitudes, stereotypes, or discrimination based on their immigrant status or nationality), marginalization, fear of deportation, and acculturation difficulties [16]. As a rapidly growing population, it is important to consider the impact of factors related to nativity status have on an individual's cancer risk and cancer disparities.

Although previous studies, such as Ellis et al. (2018), have identified racial-ethnic disparities in cancer outcomes, and Zhou et al. (2023) have observed differences in cancer outcomes by nativity status, the specific role of nativity status in the US concerning modifiable cancer risk factors remains less understood [18, 19]. Therefore, this cross-sectional epidemiological study aims to investigate the association between nativity, length of time in the US, and modifiable risk factors, considering socio-demographic characteristics.

## Methods and materials

### Ethics approval and consent to participate

This study was conducted using merged datasets from the 2010 and 2015 NHIS, approved by the Research Ethics Review Board of the National Center for Health Statistics. As the data used were publicly available, completely de-identified, and did not involve interactions with living human subjects or accessing identifiable information and does not meet the definition of human subject research. Therefore, no additional ethics approval or consent to participate was required for our analyses.

## Data source and sample

Our study utilized merged 2010 and 2015 National Health Interview Survey (NHIS) datasets as these selected years included additional questions regarding cancer prevention and control from Cancer Control Supplement. The datasets were obtained via the IPUMS Health Surveys website, and dietary variables from National Cancer Institute (NCI) Epidemiology and Genomics Research Program website. These dietary variables were already computed using algorithms developed by their staff [20, 21]. These algorithms were specifically designed to convert screener responses from the NHIS dietary questionnaire into quantitative estimates of individual dietary intake. The method of estimation relied on predictive modeling, correlating NHIS screener responses with the detailed dietary data from NHANES. This correlation ensured a robust and validated approach to dietary intake assessment, leveraging NHANES as the reference standard for dietary information [20, 21]. The key dietary variables analyzed included fruits, vegetables, dairy (cup equivalents), added sugars (teaspoon equivalents), whole grains (ounce equivalents), and fiber (grams) as guided by the ACS Guideline for Diet and Physical Activity for Cancer Prevention [20, 21]. The NHIS is a data collection program conducted by the National Center for Health Statistics (NCHS), which is part of the US Centers for Disease Control and Prevention (CDC) [22]. The NHIS collects data through continuous face-to-face household interviews and covers a wide range of health topics, with the survey content and structure periodically redesigned to meet evolving data needs [22]. The Research Ethics Review Board of the National Center for Health Statistics approved these NHIS protocols [22]. Our analyses were composed of adult respondents (aged 18 years and older) with a response for all variables included in our analyses of nativity and length of residence in the US (N = 8,861, unweighted).

## Study variables

The main dependent variables were modifiable risk factors associated with cancer including physical activity, diet, alcohol consumption, smoking status, and body mass index (BMI). The main independent variable was US nativity. Following methods from prior studies using NHIS data, adults born in the US were defined as those originating from any of the 50 states, the District of Columbia, or a US territory and we categorized adults born outside the US and its territories as foreign-born [23–27]. Foreign-born individuals were further categorized as immigrants residing in the US for <15 years (recent immigrants), and ≥15 years (long-term immigrants), this endpoint is consistent with prior research using the NHIS dataset to measure nativity and length of time in the US as a proxy for time needed to see maximal effects of acculturation [26, 27]. In this study we used dietary, exercise, smoking status, alcohol consumption, BMI, and sociodemographic characteristics (age, sex, race-ethnicity, educational attainment, employment status, annual household income) variables. Details on operationalization of all the variables included in this study are depicted in Table 1.

The ACS Guideline for Diet and Physical Activity for Cancer Prevention is structured into four principal domains [11]. The first domain is achieving and maintaining a healthy weight throughout life, which we operationalized in our study by defining adherence as a normal weight (measured as a BMI between 18.5 and 24.9). The second domain is physical activity, recommending adults engage in 150–300 minutes of moderate-intensity or 75–150 minutes of vigorous-intensity exercise weekly. Our study interpreted adherence to this domain as obtaining a minimum of 150 minutes of moderate-to-vigorous physical activity (MVPA) per week. The third domain pertains to dietary habits, promoting a healthy eating pattern across all ages. This includes a diet rich in nutrients that aid in maintaining a healthy body weight, with an emphasis on a variety of vegetables, fruits, and whole grains, and limiting or avoiding red and

**Table 1. NHIS variable assessment, exposure, and outcome operationalization.**

| Variable | Label | NHIS Question or Description | Recoding and Operationalization in Study | Derivation Details |
|---|---|---|---|---|
| RACE | Race-Ethnicity | "What race(s) do you consider yourself to be?" and "Are you Hispanic or Latino?" RACENEW provides self-reported race information using the 1997 OMB standards for race and ethnicity classification. | 1 = "Non-Hispanic White", 2 = "Non-Hispanic Black", 3 = "Hispanic", 4 = "Afro-Latin", 5 = "Non-Hispanic Asian" | Derived from RACENEW and HISPYN |
| SEX | Sex | "Are you male or female?" SEX indicates the person's sex as male or female, with options for "Unknown-refused" and "Unknown-don't know" starting in 2019. | 1 = "Male", 2 = "Female" | Direct use of NHIS SEX variable |
| BMICAT | BMI Category | Calculated from respondent's self-reported height and weight. BMICALC reports the Body Mass Index, a measure of body fat based on height and weight, calculated using the formula [Weight in pounds/ (Height in inches, squared)] multiplied by 703. | 1 = "Normal Weight", 2 = "Overweight/ Obese" | Derived from BMI values |
| YRSINUSG | Years in the US | "How long have you been living in the United States?" YRSINUSG indicates how long they have been living in the United States, with responses converted to a number of years present. | 1 = "Less than 15 years", 2 = "≥15 years" | Derived from length of residence for immigrants |
| USBORN | US Birth Status | "Were you born in the United States?" USBORN indicates whether the respondent was born in the United States, distinguishing between birth in a US state, US territory, or outside the US. | 1 = "US-Born", 2 = "Non-US-Born" | Determined based on respondent's birthplace |
| IMMIGRANT | Immigrant Status | Combination of birthplace and years in the US questions. | 1 = "Immigrant <15 Years", 2 = "Immigrant 15+ Years", 3 = "US-Born" | Derived from USBORN and YRSINUSG |
| REGIONBR | Birth Region | "In which country were you born?" REGIONBR reports region of birth by recoding the reported country of origin into one of twelve categories. | Categorized into 8 global regions | Classified based on reported country of birth |
| SMOKESTATUS2 | Smoking Status | "Have you smoked at least 100 cigarettes in your entire life?" SMOKESTATUS2 indicates the respondent's current smoking status in detailed categories of current and former smokers. | 1 = "Ever Smoker", 2 = "Never Smoker" | Classified based on smoking history |
| ALCSTAT1 | Alcohol Consumption Status | "Have you ever had at least one drink of any alcoholic beverage?" ALCSTAT1 classifies the respondent's lifetime alcohol drinking status based on their response to whether they have had at least 12 drinks of any type of alcoholic beverage in their entire life. | 1 = "Never Drinker", 2 = "Ever Drinker" | Based on lifetime alcohol consumption |
| EDUC | Educational Attainment | "What is the highest grade or level of school you have completed?" EDUC reports the highest level of schooling completed, categorized into recognized degrees and completed grades. | Categorized into four levels | Grouping based on highest education level |
| EMPSTAT | Employment Status | "Are you currently…?" (employed, unemployed, etc.) EMPSTAT reports whether the respondent was working or not in the last week, including various employment status categories. | 1 = "Employed", 2 = "Unemployed" | Based on current work status |
| AGEG | Age Group | "How old are you?" AGE reports the individual's age in years since their last birthday. | 1 = "18–34 years old", 2 = "35–54 years old", 3 = "55+ years old" | Variable Age grouped into categories |
| MARSTCOHAB | Marital/ Cohabitation Status | "What is your marital status?" MARSTCOHAB reports the person's marital status, including living with a partner. | 1 = "Married/Living Together", 2 = "Not Married or Not Living Together" | Marital status combined with cohabitation status |
| EARNINGS | Annual Earnings | "Last year, what was your total income from all sources?" EARNINGS reports the individual's total earnings during the previous calendar year. | Categorized into four groups based on income range | Income categorized into ranges |
| HISPYN | Hispanic or Not | "Are you Hispanic or Latino?" HISPYN indicates if the respondent considers themselves Hispanic or Latino. | 1 = "Yes", 2 = "No" | Based on Hispanic identification |
| HINOTCOVE | Health Insurance Coverage | "Are you currently covered by any health insurance or some other kind of healthcare plan?" HINOTCOVE indicates whether the person currently lacks health insurance coverage. | 1 = "Yes", 2 = "No" | Derived from original HINOTCOVE NHIS variable |

*(Continued)*

**Table 1.** (Continued)

| Variable | Label | NHIS Question or Description | Recoding and Operationalization in Study | Derivation Details |
|---|---|---|---|---|
| FCE | Fruit Cup Equivalents per Day | Predicted intake of fruits measured in cup equivalents per day. This variable is derived from responses to dietary screeners focusing on fruit consumption. | Quantified estimates of individual fruit intake | Derived from dietary screeners responses |
| VLNF | Vegetable Cup Equivalents per Day | Predicted intake of vegetables including legumes but excluding French fries measured in cup equivalents per day. Derived from dietary screener questions specifically excluding fries from vegetable count. | Quantified estimates of individual vegetable intake excluding fries | Derived from dietary screeners responses |
| Fiber | Fiber Intake per Day | Predicted intake of fiber measured in grams per day. This variable is estimated from dietary screeners focusing on consumption of high-fiber foods. | Quantified estimates of individual fiber intake | Derived from dietary screeners responses |
| Dairy | Dairy Cup Equivalents per Day | Predicted intake of dairy products measured in cup equivalents per day. Estimated from responses to dietary screeners related to dairy consumption. | Quantified estimates of individual dairy intake | Derived from dietary screeners responses |
| Sugar | Added Sugars from Beverages | Predicted intake of added sugars from sugar-sweetened beverages measured in teaspoon equivalents per day. Estimated based on responses to dietary screeners on consumption of sugar-sweetened beverages. | Quantified estimates of added sugars intake from beverages | Derived from dietary screeners responses |
| WHGrain | Whole Grain Ounce Equivalents | Predicted intake of whole grains measured in ounce equivalents per day. Derived from dietary screeners targeting consumption of whole grain products. | Quantified estimates of individual whole grain intake | Derived from dietary screeners responses |

processed meats, sugar-sweetened beverages, and highly processed foods. In our analyses for dietary assessment, we analyzed the consumption patterns of various food groups, including fruits, vegetables, whole grains, fiber and added sugars, in the context of the guidelines for diet and physical activity for cancer prevention set forth by the ACS [11]. In our analysis, adherence with this domain was assessed by daily consumption of fruits, vegetables, fiber, whole grains, and added sugars—measured by consumption of at least 2 cups of fruits, 2.5 cups of vegetables, 30 grams of fiber, 3 ounces of whole grains, and less than 12 tsp of added sugars per day. Lastly, the fourth domain advises avoiding or limiting alcohol consumption, categorized in our study as "ever" versus "never" drinkers. We also compared "ever" and "never" smokers to assess smoking status, considering its known association with cancer.

## Statistical analyses

All analyses conducted accounted for the NHIS complex survey design by applying appropriate sampling weights provided by NHIS. All statistical tests were two-tailed, with an alpha <0.05 considered statistically significant, and SAS version 9.4 (SAS Institute, Inc., Cary, NC, US) was used for statistical analysis. Each model was fitted using stepwise selection and Variance Inflation Factor (VIF) was assessed testing multicollinearity of variables. The results from the stepwise selection analyses indicated diverse variables that were retained for each model. Therefore, we employed a systematic approach in selecting variables uniformly across all adjusted multivariable models. Key variables such as region of birth, length of time in the US, annual income, race-ethnicity, gender, educational attainment, age, and health insurance coverage were used in all models to maintain consistency. This choice was guided by the study aims, theoretical considerations relevant to public health research, and methods employed by previous studies on nativity and length of time in the US. The robustness and appropriateness of this strategy were affirmed by the significant results (p-values <0.05) from F-tests and Wald Chi-Square tests across all models for all retained variables, along with numerous significant

associations observed in each model [23, 25–28]. The absence of multicollinearity was confirmed by the VIF values (all <2.5), further validating reliability of the models. There were no missing variables for all analyses conducted in this study. We used survey year variables from NHIS datasets in our multivariable analyses for both survey cycles. This adjustment helped account for varying impacts of survey years on dependent variables between 2010 and 2015, reducing confounding effects caused by temporal changes in our outcome variables.

Descriptive statistics and chi-squared tests were employed to describe the sociodemographic characteristics and modifiable risk factors of the sample, stratified by nativity and length of residence. Additionally, sample means with confidence limits for continuous variables (age, daily dietary intake, BMI, and weekly minutes of moderate, vigorous, and moderate-to-vigorous physical activity) were computed and were compared between groups.

Univariate and multivariable binary logistic regression models were used to examine the impact of nativity with and without controlling for sociodemographic characteristics on modifiable risk factors based on the ACS cancer prevention guidelines [11]. The dependent variables for each individual model were BMI, physical activity, smoking status, alcohol consumption, and dietary consumption patterns (fruit, vegetable, fiber, whole grains, and added sugar consumption). Independent variables included nativity and length of residence in the US, race-ethnicity (Black Hispanic, White Hispanic, non-Hispanic Black, non-Hispanic Asian, and non-Hispanic White), age groups (18–34, 35–54, and 55+ years), educational attainment (less than high school education, high school education, some college, and college graduate), sex (male and female), health insurance status (yes or no), and annual family earning brackets (<$25,00, $25,000 to $55,000, 55,000 to $74,999, and $75,000+).

## Results

### Sample description

**Distribution of sociodemographic and cancer risk factors by nativity in the US (NHIS 2010 and 2015, N = 8,861, unweighted).** As depicted in Table 2 of the total sample (N = 8,861, unweighted), recent immigrants (59.1%) were in the younger age bracket (18–34 years) compared to US-born individuals (49.0%) and long-term immigrants (27.0%). In the 35–54 years category, long-term immigrants (55.8%) outnumbered both recent immigrants (38.7%) and US-born individuals (38.4%). Among those aged 55 years and older, long-term immigrants (17.2%) and US-born individuals (12.6%) had a higher representation than recent immigrants (2.2%).

Regarding race-ethnicity, most of US-born respondents were non-Hispanic White (74.8%), while White Hispanics composed the majority among recent (56.2%) and long-term immigrants (57.1%). Non-Hispanic Asians were more prevalent among recent (21.4%) and long-term immigrants (17.2%) compared to US-born individuals (1.6%).

When examining the region of birth by immigrant status and duration in the United States, the data revealed distinct patterns among US-born individuals, recent immigrants (those with less than 15 years in the U.S.), and long-term immigrants (those with 15 or more years in the U.S.). Unsurprisingly, all US-born respondents (100%) originated from the United States. Among immigrants, those from Mexico, Central America, and the Caribbean Islands constituted the majority, with 57.0% in the recent immigrant category and 61.6% in the long-term immigrant category. This was followed by immigrants from Asia, who accounted for 21.4% of recent immigrants and 18.0% of long-term immigrants, reflecting a significant presence in the U.S. immigrant population. Notably, South America, Europe, Russia (and former USSR areas), Africa, and the Middle East also contributed to the U.S. immigrant landscape, albeit in smaller proportions. For instance, South American-born individuals represented 6.6% of

**Table 2. Distribution of sociodemographic and cancer risk factors by nativity in the US (NHIS 2010 and 2015, N = 8,861, unweighted).**

| Characteristics | US-Born (%) | Immigrant: <15 years in the US (%) | Immigrant: ≥15 Years in the US (%) |
|---|---|---|---|
| **n** | **7,370** | **563** | **928** |
| **Age***** | | | |
| 18–34 years | 49.0 | 59.1 | 27.0 |
| 35–54 years | 38.4 | 38.7 | 55.8 |
| 55+ years | 12.6 | 2.2 | 17.2 |
| **Region of Birth** | | | |
| United States | 100.0 | 0.0 | 0.0 |
| Mexico, Central America, Caribbean Islands | 0.0 | 57.0 | 61.6 |
| South America | 0.0 | 6.6 | 4.8 |
| Europe | 0.0 | 3.9 | 8.4 |
| Russia (and former USSR areas) | 0.0 | 0.6 | 1.3 |
| Africa | 0.0 | 8.6 | 4.1 |
| Middle East | 0.0 | 1.9 | 1.9 |
| Asia | 0.0 | 21.4 | 18.0 |
| **Race-ethnicity***** | | | |
| Non-Hispanic White | 74.8 | 8.2 | 13.0 |
| Non-Hispanic Black | 14.8 | 12.3 | 11.6 |
| White Hispanic | 8.3 | 56.2 | 57.1 |
| Black Hispanic | 0.5 | 1.9 | 1.2 |
| Non-Hispanic Asian | 1.6 | 21.4 | 17.2 |
| **Sex***** | | | |
| Male | 58.3 | 69.6 | 67.0 |
| Female | 41.8 | 30.4 | 33.1 |
| **Education***** | | | |
| <High school education | 6.6 | 27.9 | 28.1 |
| High school education | 25.1 | 24.9 | 19.9 |
| Some College | 37.3 | 17.6 | 23.2 |
| College graduate | 31.0 | 29.6 | 28.8 |
| **Annual income***** | | | |
| <$25,000 | 40.1 | 55.9 | 39.3 |
| $25,000 to $55,000 | 35.6 | 27.7 | 37.8 |
| 55,000 to $74,999 | 10.9 | 5.6 | 10.2 |
| $75,000+ | 13.5 | 10.8 | 12.7 |
| **Health insurance coverage***** | | | |
| Yes | 15.3 | 44.1 | 25.6 |
| No | 84.7 | 55.9 | 74.4 |
| **Physical Activity***** | | | |
| Meets criteria | 55.0 | 42.7 | 46.5 |
| Does not meet criteria | 45.0 | 57.3 | 53.5 |
| **Alcohol use***** | | | |
| Never drinker | 11.5 | 31.1 | 17.8 |
| Ever drinker | 88.5 | 68.9 | 82.2 |
| **Smoking status***** | | | |
| Ever smoker | 40.1 | 22.1 | 30.3 |
| Never smoker | 59.9 | 77.9 | 69.7 |
| **Overweight/Obesity** | | | |
| Normal Weight | 34.8 | 37.1 | 30.7 |

*(Continued)*

**Table 2.** (Continued)

| Characteristics | US-Born (%) | Immigrant: <15 years in the US (%) | Immigrant: ≥15 Years in the US (%) |
|---|---|---|---|
| **n** | **7,370** | **563** | **928** |
| Overweight/Obesity | 65.2 | 62.9 | 69.3 |
| **Fruits*** | | | |
| Meets criteria | 77.0 | 86.3 | 84.4 |
| Does not meet criteria | 23.0 | 13.7 | 15.6 |
| **Vegetables*** | | | |
| Meets criteria | 1.4 | 1.8 | 4.0 |
| Does not meet criteria | 98.6 | 98.2 | 96.0 |
| **Whole grains** | | | |
| Meets criteria | 3.9 | 5.0 | 4.0 |
| Does not meet criteria | 96.1 | 95.0 | 96.0 |
| **Added sugars** | | | |
| Meets criteria | 5.5 | 4.3 | 6.6 |
| Does not meet criteria | 94.5 | 95.7 | 93.5 |
| **Fiber*** | | | |
| Meets criteria | 2.5 | 4.9 | 5.0 |
| Does not meet criteria | 97.6 | 95.1 | 95.0 |

Notes:

* $p \leq 0.05$;

** $p \leq 0.01$;

*** $p \leq 0.0001$—significantly different among by nativity

p-values are calculated based on Chi-squared test with Rao & Scott's second-order correction

recent immigrants and 4.8% of long-term immigrants. European-born immigrants were more prevalent among long-term residents (8.4%) compared to recent arrivals (3.9%), suggesting earlier waves of migration from Europe. Similarly, immigrants from Russia and former USSR areas, though small in percentage (0.6% recent, 1.3% long-term), added to the diversity of the immigrant population. African and Middle Eastern immigrants, while constituting a minor portion (African: 8.6% recent, 4.1% long-term; Middle Eastern: 1.9% for both recent and long-term), further illustrated the wide-ranging origins of the U.S. immigrant populace.

In terms of gender, recent immigrants had a higher proportion of males (69.6%) compared to US-born (58.3%) and long-term immigrants (67.0%). Females were more prevalent among US-born (41.8%) and long-term immigrants (33.1%) than recent immigrants (30.4%).

Education levels varied. Recent immigrants had the highest percentage of individuals with less than high school education (27.9%), followed by long-term immigrants (28.1%) and US-born individuals (6.6%). College graduates were evenly distributed across all groups, with US-born individuals (31.0%) slightly leading.

Annual income differences were present, with a higher proportion of recent immigrants (55.9%) earning less than $25,000 compared to US-born individuals (40.1%). In contrast, a larger percentage of US-born individuals (13.5%) earned over $75,000 compared to recent (10.8%) and long-term immigrants (12.7%).

Health insurance coverage was significantly lower among recent immigrants (55.9% without coverage) compared to US-born (84.7% with coverage) and long-term immigrants (74.4% with coverage).

Physical activity levels showed that a lower percentage of recent immigrants (42.7%) met the criteria compared to US-born individuals (55.0%) and long-term immigrants (46.5%).

Recent immigrants had the highest proportion of never-drinkers (31.1%) compared to US-born individuals (11.5%) and long-term immigrants (17.8%). Conversely, US-born participants had the highest proportion of ever-drinkers (88.5%).

US-born individuals had the highest percentage of ever-smokers (40.1%), while recent immigrants had the lowest (22.1%).

Overweight/obesity prevalence was highest among long-term immigrants (69.3%) and US-born individuals (65.2%), compared to recent immigrants (62.9%).

Dietary factors such as fruit, vegetable, whole grain, added sugars, and fiber intake showed minimal differences across groups, except for fruit consumption where recent (86.3%) and long-term immigrants (84.4%) met the criteria more than US-born individuals (77.0%).

Rao-Scott Chi-Square tests confirmed significant associations ($< 0.001$) across all sociodemographic characteristics and cancer-related risk factor variables, besides whole grains ($p = 0.58$), added sugars ($p = 0.24$), and BMI ($p = 0.08$).

**Mean age, daily dietary intake, weekly minutes of physical activity, and body mass index (BMI) by nativity and length of residence in the US, NHIS 2010 and 2015 (N = 8,861, unweighted).** When assessing sample means from Table 3, US-born individuals had a mean age of 37.24 years (95% CI: 36.78–37.69), which was higher than the immigrants who had been in the US for less than 15 years, with a mean age of 33.17 years (95% CI: 32.18–34.16), but lower than those who had been there for $\geq$15 years, with a mean age of 42.38 years (95% CI: 41.41–43.35).

In terms of daily dietary intake, US-born individuals consumed fruits at a mean of 1.11 cups/day (95% CI: 1.09–1.13) and vegetables at 1.63 cups/day (95% CI: 1.61–1.65). Immigrants who had been in the US for less than 15 years consumed more fruits, at 1.37 cups/day (95% CI: 1.27–1.47), and vegetables, at 1.79 cups/day (95% CI: 1.70–1.88). Immigrants residing in the US for $\geq$15 years consumed fruits at 1.27 cups/day (95% CI: 1.20–1.33) and vegetables at 1.78 cups/day (95% CI: 1.72–1.84). For fiber intake, US-born participants consumed an average of

**Table 3. Mean age, daily dietary intake, weekly minutes of physical activity, and body mass index (BMI) by nativity and length of residence in the US, NHIS 2010 and 2015 (N = 8,861, unweighted).**

| Characteristic | US-Born [Mean (95% CI)] | Immigrant: <15 years in the US [Mean (95% CI)] | Immigrant: ≥15 Years in the US [Mean (95% CI)] |
|---|---|---|---|
| n | 7,370 | 563 | 928 |
| Age | 37.24 (36.78–37.69) | 33.17 (32.18–34.16) | 42.38 (41.41–43.35) |
| Fruits (cups) per day | 1.11 (1.09–1.13) | 1.37 (1.27–1.47) | 1.27 (1.20–1.33) |
| Vegetables (cups) per day | 1.63 (1.61–1.65) | 1.79 (1.70–1.88) | 1.78 (1.72–1.84) |
| Fiber (g) per day | 17.07 (16.90–17.25) | 19.88 (18.89–20.86) | 19.10 (18.41–19.78) |
| Added sugars (tsp) per day | 35.25 (34.49–36.00) | 36.61 (34.18–39.05) | 31.52 (29.60–33.43) |
| Whole grains (ounces) per day | 1.01 (0.96–1.05) | 1.00 (0.84–1.17) | 1.05 (0.88–1.21) |
| Moderate physical activity (min) per week | 120.71 (113.45–127.98) | 67.10 (52.77–81.44) | 93.16 (76.88–109.44) |
| Vigorous physical activity (min) per week | 110.87 (104.70–117.04) | 91.32 (71.37–111.27) | 81.01 (71.86–90.17) |
| MVPA (min) per week | 342.45 (326.49–358.41) | 249.74 (206.45–293.03) | 255.19 (228.35–282.03) |
| BMI (kg/m$^2$) | 28.22 (28.03–28.41) | 26.93 (26.45–27.40) | 28.00 (27.59–28.41) |

Notes:

Abbreviations: BMI- Body Mass Index, MVPA- Moderate-to-vigorous physical activity, NHIS- National Health Interview Survey

17.07 grams/day (95% CI: 16.90–17.25), compared to 19.88 grams/day (95% CI: 18.89–20.86) for recent immigrants (<15 years in the US), and 19.10 grams/day (95% CI: 18.41–19.78) for long-term immigrants (≥15 years in the US). The intake of added sugars was 35.25 teaspoons/day (95% CI: 34.49–36.00) for US-born individuals, 36.61 teaspoons/day (95% CI: 34.18–39.05) for recent immigrants, and 31.52 teaspoons/day (95% CI: 29.60–33.43) for long-term immigrants.

Physical activity level means showed that US-born individuals engaged in moderate physical activity for 120.71 minutes/week (95% CI: 113.45–127.98), vigorous activity for 110.87 minutes/week (95% CI: 104.70–117.04), and overall average MVPA of 342.45 minutes/week (95% CI: 326.49–358.41). Recent immigrants participated in moderate activity for 67.10 minutes/week (95% CI: 52.77–81.44), vigorous activity for 91.32 minutes/week (95% CI: 71.37–111.27), and overall average MVPA of 249.74 minutes/week (95% CI: 206.45–293.03). While long-term immigrants engaged in moderate activity for 93.16 minutes/week (95% CI: 76.88–109.44), vigorous activity for 81.01 minutes/week (95% CI: 71.86–90.17), and an overall average MVPA of 255.19 minutes/week (95% CI: 228.35–282.03).

Lastly, the mean BMI was 28.22 kg/m$^2$ (95% CI: 28.03–28.41) for US-born individuals, 26.93 kg/m$^2$ (95% CI: 26.45–27.40) for recent immigrants, and 28.00 kg/m$^2$ (95% CI: 27.59–28.41) for long-term immigrants.

**Univariate and binary multivariable logistic regression analyses.** The results of the binary multivariate analyses showed differences based on immigrant status by ACS adherence variables (Table 4).

Per Fig 1, when assessing BMI, recent immigrants had lower odds of being overweight/obese compared to US-born individuals in the univariate model, but this was not statistically significant (OR: 0.90, 95% CI: 0.72–1.13). In the multivariate model, the findings were not statistically significant (AOR: 0.80, 95% CI: 0.54–1.18). Long-term immigrants showed no significant difference in either the univariate (OR: 1.20, 95% CI: 1.00–1.45) or multivariate models (AOR: 0.87, 95% CI: 0.61–1.26).

Regarding physical activity, both recent and long-term immigrants had lower odds of meeting the guidelines compared to US-born individuals in the univariate model (OR: 0.61, 95% CI: 0.49–0.76 for recent immigrants; OR: 0.71, 95% CI: 0.60–0.85 for long-term immigrants). However, these differences were not statistically significant in the multivariate model for either recent (AOR: 0.76, 95% CI: 0.53–1.08) or long-term immigrants (AOR: 0.85, 95% CI: 0.62–1.17).

Alcohol consumption was notably lower among immigrants. Recent immigrants had significantly lower odds of being ever drinkers in both univariate (OR: 0.29, 95% CI: 0.23–0.37) and multivariate models (AOR: 0.33, 95% CI: 0.21–0.50). Long-term immigrants also had lower odds in both models (Univariate OR: 0.60, 95% CI: 0.48–0.75; Multivariate AOR: 0.56, 95% CI: 0.37–0.84).

In terms of smoking status, both recent and long-term immigrants had lower odds compared to US-born individuals in both univariate and multivariate models, with recent immigrants having an OR of 0.43 (95% CI: 0.33–0.55) and an AOR of 0.30 (95% CI: 0.19–0.46), and long-term immigrants having an OR of 0.65 (95% CI: 0.54–0.80) and an AOR of 0.42 (95% CI: 0.30–0.59).

For meeting fruit consumption guidelines, both recent and long-term immigrants had higher odds in both models, with recent immigrants showing a significant increase in the multivariate model (AOR: 2.80, 95% CI: 1.76–4.45).

Vegetable consumption guidelines adherence was not statistically significant for recent immigrants, but long-term immigrants had higher odds in the univariate model (OR: 2.84, 95% CI: 1.78–4.51). The findings were not statistically significant in the multivariate model for either group.

**Table 4. Univariate (OR, 95% CI) and Multivariate logistic regression models (AOR, 95% CI) for the association between US nativity/length of residence in US and non-adherence to American Cancer Society (ACS) diet and physical activity recommendations among respondents (N = 8,861, unweighted) from 2010 and 2015 National Health Interview Surveys (NHIS).**

| Adherence Variable | OR (95% CI)—Immigrant: <15 Years in US | OR (95% CI)—Immigrant: ≥15 Years in the US | AOR (95% CI)—Immigrant: <15 Years in US | AOR (95% CI)—Immigrant: ≥15 Years in the US |
|---|---|---|---|---|
| **BMI (Overweight/Obese)** [a] | 0.90 (0.72–1.13) | 1.20 (1.00–1.45) | 0.80 (0.54–1.18) | 0.87 (0.61–1.26) |
| **Physical Activity (≥150 mins MPVA/week)** [b] | 0.61 (0.49–0.76) | 0.71 (0.60–0.85) | 0.76 (0.53–1.08) | 0.85 (0.62–1.17) |
| **Alcohol Consumption (Ever Drinker)** [c] | 0.29 (0.23–0.37) | 0.60 (0.48–0.75) | 0.33 (0.21–0.50) | 0.56 (0.37–0.84) |
| **Smoking Status (Ever Smoker** [d] | 0.43 (0.33–0.55) | 0.65 (0.54–0.80) | 0.30 (0.19–0.46) | 0.42 (0.30–0.59) |
| **Meets Fruit Consumption Guidelines** [e] | 1.87 (1.35–2.60) | 1.59 (1.27–2.00) | 2.80 (1.76–4.45) | 1.87 (1.22–2.86) |
| **Meets Vegetable Consumption Guidelines** [f] | 1.27 (0.53–3.08) | 2.84 (1.78–4.51) | 0.45 (0.13–1.55) | 0.75 (0.29–1.96) |
| **Meets Fiber Consumption Guidelines** [g] | 2.09 (1.29–3.39) | 2.18 (1.41–3.39) | 2.19 (0.99–4.85) | 2.03 (1.02–4.05) |
| **Meets Added Sugar Consumption Guidelines** [h] | 0.78 (0.48–1.27) | 1.28 (0.94–1.74) | 1.40 (0.66–2.97) | 1.60 (0.89–2.86) |
| **Meets Whole Grain Consumption Guidelines** [i] | 1.34 (0.84–2.13) | 1.08 (0.70–1.67) | 2.14 (0.88–5.25) | 1.51 (0.70–3.26) |

Notes:

** US-Born served as the referent for all models.

** Adjusted models were adjusted for nativity, region of birth, length of residence in the US among immigrants, race-ethnicity, age, education level, health coverage, and income level. This adjustment is applied to all variables except the specific variable under examination in each instance of interpretation.

** Bolded values are significantly different from US-Born respondents (referent group)

Footnotes:

[a] = BMI 25+

[b] = ≥150 minutes/week

[c] = Ever drinking

[d] = Ever smoking

[e] = Less than 2 cups/day

[f] = Less than 2.5 cups/day

[g] = Less than 30 grams/day

[h] = Less than 12 tsp/day

[I] = Less than 3 ounces/day

**Abbreviations:** BMI- Body Mass Index,—Moderate-to-vigorous physical activity, NHIS- National Health Interview Survey

Both recent and long-term immigrants had higher odds of adherence to fiber consumption guidelines in the univariate model, but only long-term immigrants had significant findings in the multivariate model (AOR: 2.03, 95% CI: 1.02–4.05).

No significant differences were observed for meeting added sugar and whole grain consumption guidelines in the multivariate models for both recent and long-term immigrants. However, long-term immigrants had increased odds of meeting whole grain guidelines in the univariate model (OR: 1.08, 95% CI: 0.70–1.67).

## Discussion

### Main findings

We found nativity and length of residency in the US impacted health-related behaviors associated with cancer risks including physical activity, smoking status, alcohol consumption, and

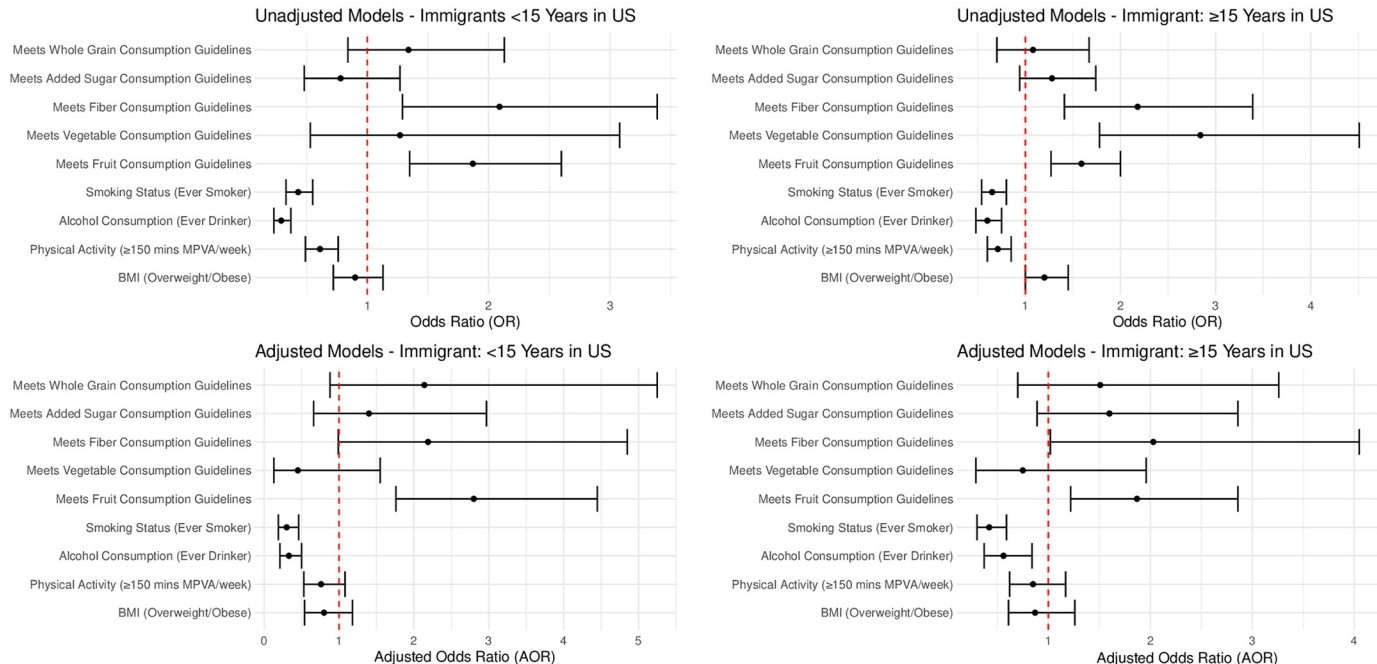

**Fig 1. Univariate (OR, 95% CI) and multivariate logistic regression models (AOR, 95% CI) for the association between US nativity/length of residence in US and non-adherence to American Cancer Society (ACS) diet and physical activity recommendations among respondents (N = 8,861, unweighted).**

dietary patterns. Prior studies have established that modifiable risk factors including physical activity, diet, alcohol consumption, smoking status, and BMI significantly influence cancer risk [11, 29–32]. Our study extends this understanding by showing that these risk factors vary among immigrants based on nativity and length of stay in the US, highlighting the need for culturally tailored cancer prevention strategies. We observed that longer US residency often leads to the adoption of local health behaviors, a phenomenon known as the "acculturation", which modifies immigrants' cancer risk profiles [33]. This may be attributed to the stark income disparities observed among recent immigrants compared to US-born individuals, as highlighted in the presented findings, underscore the critical economic challenges foreign-born US residents face.

A surprising finding in our study was the difference in health insurance coverage among nativity groups, which contrasts with past research noting a coverage gap between recent immigrants and US-born individuals. Contrary to expectations, recent immigrants (<15 years in the US) have higher health insurance coverage (44.1%) compared to US-born individuals (15.3%) and long-term immigrants (≥15 years in the US) (25.6%) [34]. This discrepancy may be due to our sample of recent immigrants benefiting from specific insurance programs or employer-based coverage, as well as their proactive efforts to secure health insurance as part of their integration process. Conversely, the high percentage of uninsured US-born individuals (84.7%) indicates significant coverage challenges that could exacerbate health disparities and negatively impact cancer prevention and outcomes [34].

A key finding is the distinct behavioral patterns among immigrants, particularly those who have resided in the US for shorter durations. These patterns include variations in dietary habits, physical activity levels, and substance use (smoking status and alcohol consumption), which collectively contribute to varying cancer risks compared to US-born individuals. We observed that recent immigrants tended to have healthier dietary patterns and lower substance

use in our models, as found in previous research, suggesting an overall protective effect against cancer risk [35]. However, this effect appears to diminish with longer residence in the US, indicating a potential impact of acculturation on health behaviors. This is consistent with findings by Gubernskaya et al, who observed an increased risk of obesity and deteriorating health among Hispanic immigrants in the US for longer periods of time [36].

The "Hispanic/Latino Paradox," originally formulated in 1986, describes the unexpected lower all-cause mortality rates experienced by Hispanics/Latinos, despite their higher poverty rates, lower educational attainment, and lower health insurance coverage rates compared to non-Hispanic Whites [37–41]. This relative advantage has been attributed to a combination of several factors and cultural norms, such as "familismo," which is a protective mechanism from unfavorable health outcomes among Hispanics compared to non-Hispanic Whites [37, 42]. However, this paradox may not uniformly apply across all Hispanic/Latino subgroups. The variance among subpopulations, such as differences in health outcomes between foreign-born and US-born Hispanics, as well as among Afro-Latinos and Afro-Caribbeans, is often overlooked. These subgroups exhibit distinct socio-economic, cultural, and health profiles, potentially leading to different health outcomes and access to healthcare services [43–47]. This was evidenced by findings from Pinheiro et al., 2011 who explored cancer survival patterns related to this paradox and found no evidence of "Hispanic advantage" found in cancer survival [42]. Notably, their findings suggested that by grouping Hispanics as a homogenous group, rather than by subgroup, existing survival disparities are missed [42]. Similarly, studies focusing on other racial-ethnic groups, such as Asians and Blacks, have also revealed significant intra-group heterogeneity, further emphasizing the need for disaggregated data in health disparities research [43, 48, 49]. This heterogeneity among racial-ethnic subpopulations has been studied among other immigrant racial-ethnic groups including Asians and Blacks and has been coined the "healthy immigrant paradox" in past studies [50–54]. Therefore, exploring inter- and intra-group differences (i.e. Black Hispanic, White Hispanic, foreign-born, US-born) across different racial-ethnic subpopulations is a crucial endeavor to truly understand the modifiable risk factors as well as behavioral characteristics to mitigate health disparities in the US [53].

### Similarities and/or differences with findings from other studies

Our results corroborate previous findings that foreign nativity influences cancer risk in the US, as demonstrated by McRoy et al. (2018) who found that immigrants had better health outcomes compared to their US-born counterpoint notwithstanding access to fewer resources, thus confirming the healthy immigrant paradox [50]. Interestingly, our study adds to the existing literature by emphasizing the role of duration of US residence in shaping cancer risk profiles. This finding aligns with Gallo et al. (2009), who noted that the health advantages associated with being non-US-born tend to wane with increased time spent in the US [55]. Further, our findings align with Miller et al. (2018), who found significant variations in cancer incidence based on nativity and length of residence, highlighting the importance of assessing the heterogeneity of the US Hispanic population beyond ethnicity [56]. Our study thus bridges a critical gap in understanding how acculturation affects health outcomes, particularly concerning cancer risk.

### Clinical and public health implications from our findings

Our findings affirm the importance of considering nativity status and length of US residence in cancer risk assessment and prevention strategies. Healthcare providers should be aware of these variations when screening for cancer risk factors [57]. Public health initiatives should

aim to develop targeted health promotion and cancer screening programs that are culturally sensitive and accessible to diverse populations, including recent immigrants.

### Strengths and limitations from our findings

Although our study provides a nuanced understanding of the interplay between nativity, duration of residence in the US, and modifiable risk factors, it has some limitations. Firstly, the reliance on self-reported cross-sectional data may introduce biases and limits our ability to establish causal relationships, and the tracking of behavioral changes over time. Further, we may not have captured all factors influencing cancer risk, such as undocumented lifestyle modifications post-immigration, cultural practices that affect health behaviors, mental health aspects which influence of stress-related factors on cancer risks among immigrant communities, and H. pylori infection (which was been shown to have significant association with foreign nativity and the incidence of gastric cancer) [57]. Moreover, we were not able to distinguish between different categories of non-US-born participants, such as temporary visa holders, permanent residents, and disaggregate the precise country of origin as countries were grouped by region. This lack of detail prevents a nuanced analysis of how varying immigration statuses might influence stress levels, resource access, and consequently, cancer risk behaviors. Additionally, the use of self-reported race is limited by its subjective nature, as individuals may interpret their racial identity differently depending on their country of origin and cultural norms [58].

Despite these limitations, our study is comprehensive due to its analysis of a large, diverse population that is largely understudied. The NHIS data set provided robust data on modifiable behaviors implicated in cancer risk and detailed information regarding nativity.

## Conclusions

### What makes our study novel

The impact of nativity and acculturation on diet, exercise, obesity, smoking, and alcohol usage in not well understood. Our analysis adds valuable contributions to the existing literature related to cancer risks and migrant US populations by presenting evidence of the changes in health behaviors among immigrants over time, emphasizing the role of acculturation in the adoption of behaviors associated with increased cancer risk particularly less fruit consumption and substance use.

### Importance and relevance of the study

This knowledge is crucial for health policymakers and practitioners who work with immigrant communities and minorities. Foreign born populations make up a significant proportion of minority subpopulations living in the US [13, 59, 60]. Therefore, there is a need for culturally tailored health promotion strategies that can address the unique challenges faced by immigrants in maintaining healthy behaviors throughout their years residing in the US such as health insurance coverage and access to health care services.

### Future directions

Future research should explore the impact these modifiable risk factors may have on cancer mortality. Additionally, longitudinal studies should be employed to track the evolution of health behaviors among immigrants and explore interventions that effectively support healthy lifestyle choices, and work to help immigrant maintain current healthy behaviors as they acculturate. Moreover, investigating the underlying factors driving these behavioral changes, such

as socioeconomic status and cultural influences, would further enhance our understanding of health risks among immigrant populations. Finally, culturally targeted considerations for lifestyle interventions where dietary choices and exercise modalities which resonate with immigrant populations and their dynamic cultures must be created to aid in effective lifestyle modification interventions for these immigrant populations.

## Supporting information

**S1 Table. Multivariate logistic regression models (AOR, 95% CI) for the association between US nativity/length of residence in US and BMI, NHIS 2010 and 2015 (N = 8,861, unweighted).**
(XLSX)

**S2 Table. Multivariate logistic regression models (AOR, 95% CI) for the association between US nativity/length of residence in US and physical activity, NHIS 2010 and 2015 (N = 8,861, unweighted).**
(XLSX)

**S3 Table. Multivariate logistic regression models (AOR, 95% CI) for the association between US nativity/length of residence in US and smoking status, NHIS 2010 and 2015 (N = 8,861, unweighted).**
(XLSX)

**S4 Table. Multivariate logistic regression models (AOR, 95% CI) for the association between US nativity/length of residence in US and alcohol consumption, NHIS 2010 and 2015 (N = 8,861, unweighted).**
(XLSX)

**S5 Table. Multivariate logistic regression models (AOR, 95% CI) for the association between US nativity/length of residence in US and fruit consumption, NHIS 2010 and 2015 (N = 8,861, unweighted).**
(XLSX)

**S6 Table. Multivariate logistic regression models (AOR, 95% CI) for the association between US nativity/length of residence in US and vegetable consumption, NHIS 2010 and 2015 (N = 8,861, unweighted).**
(XLSX)

**S7 Table. Multivariate logistic regression models (AOR, 95% CI) for the association between US nativity/length of residence in US and fiber consumption, NHIS 2010 and 2015 (N = 8,861, unweighted).**
(XLSX)

**S8 Table. Multivariate logistic regression models (AOR, 95% CI) for the association between US nativity/length of residence in US and added sugar consumption, NHIS 2010 and 2015 (N = 8,861, unweighted).**
(XLSX)

**S9 Table. Multivariate logistic regression models (AOR, 95% CI) for the association between US nativity/length of residence in US and whole grain consumption, NHIS 2010 and 2015 (N = 8,861, unweighted).**
(XLSX)

## Acknowledgments

The authors acknowledge the invaluable support and mentorship provided by the CRANE Research Lab. Special thanks are extended to members Melissa Lopez-Pentecost, Atif Bhatti, and Zuhair Khan for their initial contributions to the abstract of this study.

## Author Contributions

**Conceptualization:** LaShae D. Rolle, Tracy E. Crane.

**Data curation:** LaShae D. Rolle.

**Formal analysis:** LaShae D. Rolle, Amrit Baral.

**Methodology:** LaShae D. Rolle, Tracy E. Crane.

**Supervision:** Tracy E. Crane.

**Validation:** Tracy E. Crane.

**Visualization:** LaShae D. Rolle, Rolando F. Trejos.

**Writing – original draft:** LaShae D. Rolle, Alexa Parra, Amrit Baral, Rolando F. Trejos, Maurice J. Chery.

**Writing – review & editing:** LaShae D. Rolle, Alexa Parra, Amrit Baral, Rolando F. Trejos, Maurice J. Chery, Reanna Clavon, Tracy E. Crane.

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
