## [Decision Letter · Decision Letter 0]

23 Feb 2024

PONE-D-24-01140Differences in Cancer Risk Behaviors by Nativity (US-Born v. Non-US-Born) and Length of Time in the USPLOS ONE

Dear Dr. Rolle,

Thank you for submitting your manuscript to PLOS ONE. After careful consideration, we feel that it has merit but does not fully meet PLOS ONE’s publication criteria as it currently stands. Therefore, we invite you to submit a revised version of the manuscript that addresses the points raised during the review process.

**ACADEMIC EDITOR: **

The authors have written a well thought out paper. However the paper lacks some nuances as noted by the reviewers. This includes the following : 

Additionally, further clarity in the discussion on the "Hispanic/Latino Paradox" and the importance of considering subpopulations within racial-ethnic groups could provide a deeper understanding of health disparities.

The authors may want to review the comments and resubmit the paper for further consideration. 

We look forward to receiving your revised manuscript.

Kind regards,

Souparno Mitra, M.D.

Academic Editor

PLOS ONE

3. Please remove your figures from within your manuscript file, leaving only the individual TIFF/EPS image files, uploaded separately. These will be automatically included in the reviewers’ PDF.

Reviewers' comments:

Reviewer's Responses to Questions

**Comments to the Author**

1. Is the manuscript technically sound, and do the data support the conclusions?

Reviewer #1: Yes

Reviewer #2: Yes

2. Has the statistical analysis been performed appropriately and rigorously? 

Reviewer #1: Yes

Reviewer #2: Yes

3. Have the authors made all data underlying the findings in their manuscript fully available?

Reviewer #1: Yes

Reviewer #2: Yes

4. Is the manuscript presented in an intelligible fashion and written in standard English?

Reviewer #1: Yes

Reviewer #2: Yes

5. Review Comments to the Author

Reviewer #1: The paper examines the relationship between US nativity, length of residence in the US, and modifiable cancer risk factors present various arguments related to the quality and implications of the study

The term "social determinants of health (SDoH)" is introduced in line 66, but could be further explained for readers who may not be familiar with the concept. Providing examples of SDoH that contribute to health disparities would enhance understanding.

In lines 71-72, please provide reference to the previous research studies mentioned

Generalizability: Without diverse immigrant populations included, it's difficult to generalize the patterns observed to all immigrants, much less to different cultural or national contexts outside the US.

While the study effectively identifies distinct behavioral patterns among immigrants, a more nuanced exploration of these behaviors and their direct implications for cancer risk would enhance the overall impact of the findings. Additionally, further clarity in the discussion on the "Hispanic/Latino Paradox" and the importance of considering subpopulations within racial-ethnic groups could provide a deeper understanding of health disparities.

Limited Information on Immigration Status: The lack of detail on the immigration status of non-US-born participants limits the depth and nuance of the study's conclusions. For example, temporary visa holders may experience different stresses or have different access to resources compared to permanent residents, factors that could influence cancer risk behaviors.

Causality and Lack of Follow-Up: A cross-sectional design inherently limits the ability to derive causal relationships. For a stronger argument, a longitudinal approach would be beneficial, tracking changes in the behaviors of the same cohort over time.

In conclusion, this study contributes valuable insights to the field of cancer disparities, urging healthcare providers and public health initiatives to adopt culturally sensitive and targeted strategies, particularly for recent immigrants. As we appreciate the study's strengths, such as its comprehensive analysis, acknowledgment of income disparities, and recognition of the health insurance coverage gap, addressing the specified areas for improvement would further enhance the clarity and impact of the research.

Reviewer #2: This article presents an interesting research on differences in modifiable cancer risk factors based on US nativity and length of stay in the US. The study found differences in exercise levels, fruit and vegetable consumption in different groups. There were some minor issues that I could find in this study:

1. Title: It might be helpful if you specify "Modifiable" Cancer Risk Behaviors.

2. Abstract: line 23 and 25: please specify that you are talking about "modifiable risk factors" for cancers.

3. Introduction: you have very briefly touched upon the influence of nativity on cancer without describing existing literature on this. It would be better if you could elaborate this a bit more.

4. Data source and sample:

Line 78: NHIS is done every year but you chose 2010 and 2015 years for your data. Please elaborate the reason for this choice.

Line 80-82: Please elaborate the validity of this algorithm from NCI to derive the dietary intake from your screener.

5. Study Variables:

Table 1: Why were variables like calcium, sugar, SSB derived and described here? I did not see them being used elsewhere in the study.

Line 126-128: Was there a reason why you chose "ever smoked" and "ever consumed alcohol" to assess modifiable risk factors in a cross sectional design? We know that quitting smoking and alcohol use leads to reduction in cancer risks after several years. Especially when we are trying to understand differences in risk factors with length of stay in the US, a more proximate indicator of smoking and alcohol use might have been helpful (e.g. smoking in last 5 or 10 years).

6. Statistical analysis:

Line 145-146: You are using data of a national sample (NHIS), you had no direct contact with the clients. So it is not necessary to mention Informed consent.

Line 155: How did you compute the moderate to vigorous physical activity variable? The values do not seem to be a mean of moderate and vigorous physical activity variables in Table 3.

7. Results:

Table 2: You seem to have erroneously flipped the Health Insurance Coverage categories (Yes and No)

Figure 1: It has several typographical errors. For example, in Unadjusted Models: Immigrants <15 years: "Meets Fiber Consumption Guidelines Meets", next line add "meets" before Fruit Consumption Guidelines. There are similar errors in all four categories. Also, it would look better if the variables are arranged in similar order in all 4 categories in this figure.

8. Conclusions: Line 379: change "fruit consumption" to "less fruit consumption".

6. PLOS authors have the option to publish the peer review history of their article (what does this mean?). If published, this will include your full peer review and any attached files.

Reviewer #1: **Yes: **VINOD SHARMA

Reviewer #2: No

---

## [Author Response · Author response to Decision Letter 0]

29 Feb 2024

Dear Editor,

Thank you for the thoughtful review of our manuscript. We deeply appreciate the insights and suggestions provided. Below, we have outlined our responses (underlined and bolded) and detailed the changes we've made to the manuscript. These changes (highlighted in yellow) are in the revised document.

REVIEWER: 1

Review Comments to the Author

Reviewer #1: The paper examines the relationship between US nativity, length of residence in the US, and modifiable cancer risk factors present various arguments related to the quality and implications of the study.

1. The term "social determinants of health (SDoH)" is introduced in line 66 but could be further explained for readers who may not be familiar with the concept. Providing examples of SDoH that contribute to health disparities would enhance understanding.

Thank you for your suggestion to expand on the concept of "social determinants of health (SDoH)" for readers who may not be familiar with it. In response, we have added a brief explanation and examples of SDoH in the context of our study, which focuses on the health disparities faced by immigrants in the US.

Foreign-born individuals in the US face health disparities related to social determinants of health (SDoH). SDoH are conditions in the environments where people are born, live, learn, work, play, and age that affect a wide range of health and quality of life outcomes and risks (15). Examples of SDoH contributing to health disparities in immigrants include limited access to healthcare, socioeconomic status, language barriers, and living conditions (16,17). These are often compounded by stigma (negative attitudes, stereotypes, or discrimination based on their immigrant status or nationality), marginalization, fear of deportation, and acculturation difficulties (16).

2. In lines 71-72, please provide reference to the previous research studies mentioned.

This request has been completed.

Although previous studies, such as Ellis et al. (2018), have identified racial-ethnic disparities in cancer outcomes, and Zhou et al. (2023) have observed differences in cancer outcomes by nativity status, the specific role of nativity status in the US concerning modifiable cancer risk factors remains less understood (16,17).

3. Generalizability: Without diverse immigrant populations included, it's difficult to generalize the patterns observed to all immigrants, much less to different cultural or national contexts outside the US.

Thank you for your comments regarding the generalizability of our study findings to diverse immigrant populations. In response to your concerns, we have enhanced our analysis to include a more detailed breakdown of the region of birth by immigrant status and duration in the United States in the descriptive table. This enhancement, as outlined in the revised manuscript, provides a clearer and more comprehensive view of the diversity within the immigrant population in our study. The new data distinctly categorizes US-born individuals, recent immigrants (those with less than 15 years in the U.S.), and long-term immigrants (those with 15 or more years in the U.S.), and further delineates their regions of origin. This allows for a nuanced understanding of the immigrant landscape in the U.S., showcasing a wide array of regions including Mexico, Central America, the Caribbean Islands, South America, Europe, Russia and former USSR areas, Africa, the Middle East, and Asia. The inclusion of these diverse regions of birth, particularly with the differentiation between recent and long-term immigrants, significantly enhances the generalizability of our findings. It provides a more accurate reflection of the U.S. immigrant population's diversity, thereby addressing the concern about the generalizability of our study to various immigrant groups. We believe that this detailed categorization and analysis effectively address your concerns about the representation of diverse immigrant populations in our study. The added data allows for a more inclusive understanding of cancer risk behaviors and health disparities across different immigrant groups in the U.S., strengthening the overall impact and relevance of our findings. We thank you for this valuable suggestion!

When examining the region of birth by immigrant status and duration in the United States, the data revealed distinct patterns among US-born individuals, recent immigrants (those with less than 15 years in the U.S.), and long-term immigrants (those with 15 or more years in the U.S.). Unsurprisingly, all US-born respondents (100%) originated from the United States. Among immigrants, those from Mexico, Central America, and the Caribbean Islands constituted the majority, with 57.0% in the recent immigrant category and 61.6% in the long-term immigrant category. This was followed by immigrants from Asia, who accounted for 21.4% of recent immigrants and 18.0% of long-term immigrants, reflecting a significant presence in the U.S. immigrant population. Notably, South America, Europe, Russia (and former USSR areas), Africa, and the Middle East also contributed to the U.S. immigrant landscape, albeit in smaller proportions. For instance, South American-born individuals represented 6.6% of recent immigrants and 4.8% of long-term immigrants. European-born immigrants were more prevalent among long-term residents (8.4%) compared to recent arrivals (3.9%), suggesting earlier waves of migration from Europe. Similarly, immigrants from Russia and former USSR areas, though small in percentage (0.6% recent, 1.3% long-term), added to the diversity of the immigrant population. African and Middle Eastern immigrants, while constituting a minor portion (African: 8.6% recent, 4.1% long-term; Middle Eastern: 1.9% for both recent and long-term), further illustrated the wide-ranging origins of the U.S. immigrant populace.

Table 2. Distribution of Sociodemographic and Cancer Risk Factors by Nativity in the US (NHIS 2010 and 2015, N=8,861, unweighted).

Characteristics US-Born (%) Immigrant: <15 years in the US (%) Immigrant: ≥15 Years in the US (%)

Region of Birth 

United States 100.0 0.0 0.0

Mexico, Central America, Caribbean Islands 0.0 57.0 61.6

South America 0.0 6.6 4.8

Europe 0.0 3.9 8.4

Russia (and former USSR areas) 0.0 0.6 1.3

Africa 0.0 8.6 4.1

Middle East 0.0 1.9 1.9

Asia 0.0 21.4 18.0

4. While the study effectively identifies distinct behavioral patterns among immigrants, a more nuanced exploration of these behaviors and their direct implications for cancer risk would enhance the overall impact of the findings. 

Thank you for your insightful feedback. In response, we have added a concise section to our discussion highlighting how modifiable risk factors such as physical activity, diet, alcohol consumption, smoking status, and BMI vary among immigrants based on nativity and length of stay in the US. This addition emphasizes the 'acculturation effect' and the need for culturally tailored cancer prevention strategies. We believe this enhancement addresses your suggestion by providing a clearer understanding of the relationship between immigration, acculturation, and cancer risk factors.

We found nativity and length of residency in the US impacted health-related behaviors associated with cancer risks including physical activity, smoking status, alcohol consumption, and dietary patterns. Prior studies have established that modifiable risk factors including physical activity, diet, alcohol consumption, smoking status, and BMI significantly influence cancer risk (11,29–32). Our study extends this understanding by showing that these risk factors vary among immigrants based on nativity and length of stay in the US, highlighting the need for culturally tailored cancer prevention strategies. We observed that longer US residency often leads to the adoption of local health behaviors, a phenomenon known as the “acculturation”, which modifies immigrants' cancer risk profiles (33).

5. Additionally, further clarity in the discussion on the "Hispanic/Latino Paradox" and the importance of considering subpopulations within racial-ethnic groups could provide a deeper understanding of health disparities.

Thank you for your valuable feedback. In response, we have expanded our discussion on the "Hispanic/Latino Paradox" to address the importance of considering intra-group variations among Hispanic/Latino subpopulations. This revised section now includes a more detailed analysis of the diverse socio-economic, cultural, and health profiles within these groups and parallels drawn with other racial-ethnic groups to underscore the significance of disaggregated data in health disparities research. We believe this addition provides the depth and clarity needed to enhance understanding of health disparities in diverse populations.

The “Hispanic/Latino Paradox," originally formulated in 1986, describes the unexpected lower all-cause mortality rates experienced by Hispanics/Latinos, despite their higher poverty rates, lower educational attainment, and lower health insurance coverage rates compared to non-Hispanic Whites (32–36). This relative advantage has been attributed to a combination of several factors and cultural norms, such as “familismo,” which is a protective mechanism from unfavorable health outcomes among Hispanics compared to non-Hispanic Whites (32,37). However, this paradox may not uniformly apply across all Hispanic/Latino subgroups. The variance among subpopulations, such as differences in health outcomes between foreign-born and US-born Hispanics, as well as among Afro-Latinos and Afro-Caribbeans, is often overlooked. These subgroups exhibit distinct socio-economic, cultural, and health profiles, potentially leading to different health outcomes and access to healthcare services (38–42). This was evidenced by findings from Pinheiro et al., 2011 who explored cancer survival patterns related to this paradox and found no evidence of “Hispanic advantage” found in cancer survival (37). Notably, their findings suggested that by grouping Hispanics as a homogenous group, rather than by subgroup, existing survival disparities are missed (37). Similarly, studies focusing on other racial-ethnic groups, such as Asians and Blacks, have also revealed significant intra-group heterogeneity, further emphasizing the need for disaggregated data in health disparities research (43–45). This heterogeneity among racial-ethnic subpopulations has been studied among other immigrant racial-ethnic groups including Asians and Blacks and has been coined the “healthy immigrant paradox” in past studies (46–50). Therefore, exploring inter- and intra-group differences (i.e. Black Hispanic, White Hispanic, foreign-born, US-born) across different racial-ethnic subpopulations is a crucial endeavor to truly understand the modifiable risk factors as well as behavioral characteristics to mitigate health disparities in the US (49).

6. Limited Information on Immigration Status: The lack of detail on the immigration status of non-US-born participants limits the depth and nuance of the study's conclusions. For example, temporary visa holders may experience different stresses or have different access to resources compared to permanent residents, factors that could influence cancer risk behaviors.

Thank you for your comment on the limitations stemming from the lack of detailed information on immigration status in our dataset. We acknowledge that this is an important aspect, and its absence indeed restricts the depth of our analysis. The dataset utilized, primarily sourced from the NHIS, unfortunately does not differentiate between various categories of non-US-born participants, such as temporary visa holders and permanent residents. As such, our study's scope is confined to broader comparisons between native and foreign-born individuals without delving into the nuances of different immigration statuses. We recognize the potential differences in stress levels and access to resources among these subgroups and agree that these could significantly impact cancer risk behaviors. Future studies with more detailed datasets could explore these distinctions to enrich our understanding of these complex dynamics. Despite this limitation, we believe our study provides valuable insights into the general trends in cancer risk behaviors among immigrant populations in the US.

This has been added as a limitation.

Moreover, we were not able to distinguish between different categories of non-US-born participants, such as temporary visa holders, permanent residents, and disaggregate the precise country of origin as countries were grouped by region. This lack of detail prevents a nuanced analysis of how varying immigration statuses might influence stress levels, resource access, and consequently, cancer risk behaviors.

7. Causality and Lack of Follow-Up: A cross-sectional design inherently limits the ability to derive causal relationships. For a stronger argument, a longitudinal approach would be beneficial, tracking changes in the behaviors of the same cohort over time.

Thank you for highlighting the limitations regarding causality and lack of follow-up inherent in our cross-sectional study design. We acknowledge that cross-sectional studies, by their nature, do not allow for the establishment of causality or tracking changes over time. Our study, utilizing publicly available data from specific time points (2010 and 2015), primarily aimed to identify associations and prevalence rates of certain risk factors in the population. While longitudinal studies indeed offer more robust insights into causal relationships and behavioral changes, our study provides valuable baseline data that can inform and justify the need for such future longitudinal research. The cross-sectional design was chosen due to its feasibility and the availability of data, offering an important snapshot that contributes to the broader understanding of the subject at hand. 

This has been added as a limitation.

Firstly, the reliance on self-reported cross-sectional data may introduce biases and limits our ability to establish causal relationships, and the tracking of behavioral changes over time.

In conclusion, this study contributes valuable insights to the field of cancer disparities, urging healthcare providers and public health initiatives to adopt culturally sensitive and targeted strategies, particularly for recent immigrants. As we appreciate the study's strengths, such as its comprehensive analysis, acknowledgment of income disparities, and recognition of the health insurance coverage gap, addressing the specified areas for improvement would further enhance the clarity and impact of the research.

REVIEWER: 2

Review Comments to the Author

This article presents an interesting research on differences in modifiable cancer risk factors based on US nativity and length of stay in the US. The study found differences in exercise levels, fruit and vegetable consumption in different groups. There were some minor issues that I could find in this study:

1. Title: It might be helpful if you specify "Modifiable" Cancer Risk Behaviors.

This request has been completed.

Differences in Modifiable Cancer Risk Behaviors by Nativity (US-Born v. Non-US-Born) and Length of Time in the US

2. Abstract: line 23 and 25: please specify that you are talking about "modifiable risk factors" for cancers.

This request has been completed.

Previous studies have identified racial-ethnic disparities in modifiable risk factors for cancers. However, the impact of US nativity on these risks is understudied. Hence, we assessed the association between US nativity and length of time in the US on modifiable cancer risk factors.

3. Introduction: you have very briefly touched upon the influence of nativity on cancer without describing existing literature on this. It would be better if you could elaborate this a bit more.

Thank you for your valuable comment regarding the need for a more detailed discussion of the existing literature on the influence of nativity on cancer in the introduction of our manuscript. We appreciate this opportunity to enhance the comprehensiveness of our work.

Although previous studies, such as Ellis et al. (2018), have identified racial-ethnic disparities in cancer outcomes, and Zho

---

## [Decision Letter · Decision Letter 1]

30 May 2024

Differences in Modifiable Cancer Risk Behaviors by Nativity (US-Born v. Non-US-Born) and Length of Time in the US

PONE-D-24-01140R1

Dear Dr. Rolle,

We’re pleased to inform you that your manuscript has been judged scientifically suitable for publication and will be formally accepted for publication once it meets all outstanding technical requirements.

Kind regards,

Souparno Mitra, M.D.

Academic Editor

PLOS ONE

Additional Editor Comments (optional):

Reviewers' comments:

Reviewer's Responses to Questions

**Comments to the Author**

1. If the authors have adequately addressed your comments raised in a previous round of review and you feel that this manuscript is now acceptable for publication, you may indicate that here to bypass the “Comments to the Author” section, enter your conflict of interest statement in the “Confidential to Editor” section, and submit your "Accept" recommendation.

Reviewer #2: All comments have been addressed

Reviewer #3: All comments have been addressed

2. Is the manuscript technically sound, and do the data support the conclusions?

Reviewer #2: Yes

Reviewer #3: Yes

3. Has the statistical analysis been performed appropriately and rigorously? 

Reviewer #2: Yes

Reviewer #3: Yes

4. Have the authors made all data underlying the findings in their manuscript fully available?

Reviewer #2: Yes

Reviewer #3: Yes

5. Is the manuscript presented in an intelligible fashion and written in standard English?

Reviewer #2: Yes

Reviewer #3: Yes

6. Review Comments to the Author

Reviewer #2: Thank you for revising the manuscript thoroughly. All reviewer comments have been adequately addressed. The reviewer 2's point 7 about the health insurance coverage in Table 2 still needs some elaboration in the discussion. You mentioned that in recent immigrants, lower income and poor health insurance coverage puts them at a higher risk of adverse health outcomes due to poor access to care. However, in Table 2, recent immigrants seem to have better insurance coverages than the US natives and long term immigrants. In fact, it is surprising to see that 85% US natives did not have insurance coverage. These findings can be discussed more.

Also, can you rephrase the first sentence in the Introduction section "Cancer disparities and inequities in the United States (US) have resulted in increased cancer risk, lower cancer screening rates and other preventive measures, delayed cancer diagnosis, poorer cancer-related treatment outcomes, and increased cancer-related burden". It is unclear what cancer disparities and inequalities are you talking about that is contributing to the above problems.

Reviewer #3: (No Response)

7. PLOS authors have the option to publish the peer review history of their article (what does this mean?). If published, this will include your full peer review and any attached files.

Reviewer #2: No

Reviewer #3: **Yes: **Tulasi Srinivasa Kumar Goriparthi

---

## [Editor Report · Acceptance letter]

13 Jun 2024

PONE-D-24-01140R1 

PLOS ONE

Dear Dr. Rolle, 

I'm pleased to inform you that your manuscript has been deemed suitable for publication in PLOS ONE. Congratulations! Your manuscript is now being handed over to our production team.

Kind regards, 

on behalf of

Dr. Souparno Mitra 

Academic Editor

PLOS ONE